# Ocular Adverse Events after Inactivated COVID-19 Vaccination in Xiamen

**DOI:** 10.3390/vaccines10030482

**Published:** 2022-03-21

**Authors:** Xiuju Chen, Xiaoxin Li, Haibo Li, Minghan Li, Songjian Gong

**Affiliations:** 1Xiamen Eye Center, Xiamen University, Xiamen 361005, China; joyychen@aliyun.com (X.C.); hyberlee@163.com (H.L.); doctorleemh2013@163.com (M.L.); loyalgg@163.com (S.G.); 2Fujian Key Laboratory of Ocular Surface and Corneal Diseases, Xiamen University, Xiamen 361005, China; 3People’s Hospital of Peking University, Beijing 100033, China

**Keywords:** COVID-19, inactivated vaccine, ocular adverse events, vaccine-induced uveitis

## Abstract

**Aims:** To report potential vaccine-induced ocular adverse events following inactivated COVID-19 vaccination (Sinopharm and Sinovac). **Methods:** This case series took place at a tertiary referral center in the southeast of China (Xiamen Eye Center in Fujian Province) from February 2021 to July 2021. Patients who received the first dose of inactivated COVID-19 vaccine and developed vaccine-related ocular adverse events within 10 days were included. The diagnosis of vaccine-related ocular adverse events was guided by the World Health Organization causality assessment and the Naranjo criteria. **Results:** Ten eyes of seven patients (two male individuals) presenting with ocular complaints following COVID-19 vaccine were included in the study. The mean (SD) age was 41.4 (9.3) years (range, 30–55 years). The mean time of ocular adverse event manifestations was 4.9 days (range, 1–10 days). Three patients were diagnosed with Vogt–Koyanagi–Harada (VKH)-like uveitis, one with multifocal choroiditis, one with episcleritis, one with iritis, and one with acute idiopathic maculopathy. Two patients received the second dose of vaccine. One patient had exacerbation of VKH, and one patient had no symptoms. An aqueous humor analysis in three patients revealed elevated proinflammatory cytokines and negative virus copy. All the patients had transient ocular disturbance and responded well to steroids. No recurrence was noted during 6 months of follow-up. **Conclusions:** Potential ocular adverse events should be reported to increase the awareness of the health community for timely detection and proper treatment.

## 1. Introduction

The COVID-19 pandemic is ongoing and has caused more than 247 million infections and 5 million deaths [1]. China’s CoronaVac and Sinopharm vaccines are widely used in China and account for almost half of the 7.3 billion COVID-19 vaccine doses delivered globally, and have been enormously important in fighting the pandemic, particularly in less wealthy nations since the World Health Organization (WHO) authorized them for emergency use [2]. Both inactivated COVID-19 vaccines were absorbed with aluminum-based adjuvant and have been reported tolerable and immunogenic in healthy people with two doses administered 21 days apart [3,4]. Although effective and well-tolerated in most patients, COVID-19 vaccines can uncommonly cause various ocular adverse effects [5,6,7,8,9].

As with drugs, the establishment of causality between the vaccine and an observed event needs a number of factors, most of which are a clear temporal association, lack of an identifiable alternate cause, and recurrence on re-challenge [10,11]. Based on the Naranjo scale and the WHO Adverse Drug Terminology, we hereby present a case series of potential ocular adverse events in a tertiary referral center in Xiamen (China) after administration of inactivated COVID-19 vaccines.

## 2. Material and Methods

### 2.1. Study Design and Inclusion Criteria

A retrospective case series of patients presenting at the retina and uveitis clinics developed ocular symptoms within 10 days following administration of the first dose of inactivated vaccines (Sinopharm and Sinovac). The Naranjo criteria for quantitative assessment of causation between medication and an adverse reaction (Appendix A) as well as the WHO causality assessment of suspected adverse drug reactions (Appendix B) were used to support the diagnosis of these cases as being vaccine-related [10,11]. The study was approved by the Institutional Review Board of the Xiamen Eye Center and performed in accordance with the ethical standards of the Declaration of Helsinki. Written informed consent was obtained from all enrolled individuals.

### 2.2. Data Collection

All the patients underwent Snellen’s best-corrected visual acuity and a slip-lamp biomicroscopic examination. An ultrasound B scan, color fundus photography (Carl Zeiss Clarus 500), optical coherence tomography (OCT) (Spectralis HRA OCT; Heidelberg Engineering) and fluorescein angiography (FA) were obtained on the initial and the following visits if needed. Gender, age, medication, medical and ocular histories were self-reported. Clinical data were collected, including systemic and ocular symptoms following vaccination, the time interval between vaccination and disease onset, laterality of eye disease, treatment, and outcome.

Accessory tests were performed with the aim to exclude other causes of ocular events. These included a complete blood count, biochemical analysis, erythrocyte sedimentation rate (ESR), C-reactive protein (CRP), rheumatoid factor (RF), antinuclear antibodies (ANA), and human leukocyte antigen (HLA)-B27. Serologic screening for TORCH, HIV, syphilis (treponema pallidum hemagglutination), and tuberculosis (purified protein derivative) were performed in all cases. Intraocular fluid was sent to detect infection, including COVID-19 virus, toxoplasmosis (TOXO), herpes virus (HSV), varicella-zoster virus (VZV), cytomegalovirus (CMV), and proinflammatory cytokines, including interleukin (IL)-2, IL-4, IL-6, IL-8, IL-10, IL-17, IFN-γ, IFN-α, and TNF-α (Qingdao Raisecare Biotech) if applicable.

## 3. Results

The study included 10 eyes of seven patients (two male individuals) presenting with ocular complaints following the first injection of COVID-19 vaccine. The mean (SD) age was 30.4 (14.7) years (range, 10–57 years). The median time of symptoms onset after vaccination was 4.9 days (range, 1–10 days). All patients were followed up for 6 months and no recurrence was noted.

Five out of seven patients had no history of ocular and systemic disease, of which three patients developed bilateral Vogt–Koyanagi–Harada (VKH)-like uveitis, one was diagnosed with unilateral multifocal choroiditis, and one with unilateral acute idiopathic maculopathy (AIM).

Past medical history of uveitis was remarkable in two patients. One patient with post-vaccine iritis had HLA-B27-related ankylosing spondylitis being well controlled with etanercept for 4 years (Case 1, see more information in selected cases). Another patient (Case 7) presented with episcleritis and had a history of intermediate uveitis on regular follow-up in our clinic, well controlled with adalimumab and methotrexate for 2 years. The episcleritis resolved within 1 week by topical steroids and without relapse for 6 months. Patients’ demographics, medical history, symptoms, and treatment are summarized in Table 1 and Table 2.

Only two patients received the second dose of the vaccine. One patient (Case 2, see more information in selected cases below ) who had a decreased vision 10 days after the first dose of Sinopharm vaccine, had exacerbation of vision loss to hand motion in the right eye (OD) 2 days following the second dose of Sinovac vaccine. Prior to the second dose vaccination, the patient had not received any treatment. The patient was diagnosed with bilateral VKH and treated with oral steroids. The vision recovered to 1.0 OD in a 4-week follow-up. One patient (Case 4) diagnosed with multifocal choroiditis in the left eye (OS) after the first dose of Sinovac vaccine immediately received periocular triamcinolone acetonide 40 mg OS. The patient got the second dose of Sinovac vaccine one month later without visual deterioration.

Aqueous humor was analyzed in three selected patients (Case 1 with iritis, Case 3 and Case 4 with VKH-like disease) and revealed significantly elevated IL-6 (range 734.2–2120.4 pg/mL) and mild to moderate elevated IFN-γ (range 27.4–855.0 pg/mL). IL-4 and IL-10 were within normal limits. The results of COVID-19 virus copy, CMV, HSV, VZV, and TORCH IgM were negative.

### Selected Cases

Case 1

A 30-year-old male was referred to our clinic with redness and blurred vision in the left eye 2 days after receiving an inactivated COVID-19 vaccine. The patient’s medical history was remarkable for ankylosing spondylitis, well-controlled with etanercept and free of uveitis attack for 4 years. Visual acuity on presentation was 0.4 OS with an intraocular pressure of 13 mmHg, and a slit lamp examination revealed a conjunctiva congestion, fine keratic precipitate, anterior chamber flare and iris adhesion (Figure 1A). The blood test revealed positive HLA-B27 and an elevated ESR of 38.0 mm/h. Tests for HIV, syphilis, and tuberculosis were negative. Aqueous humor was sent for viral and bacterial detection, including cytomegalovirus, rubella virus, herpes simplex virus as well as COVID-19 virus with negative results. In contrast, the inflammatory analysis demonstrated elevated IL-6 (2024.0 pg/mL) and IFN-γ (855.0 pg/mL). Iritis reactivation was diagnosed, and topical and periocular steroid was administered. The inflammation resolved in a 2-week follow-up with 1.0 vision (Figure 1B).

Case 2

A previously healthy 57-year-old female presented to our clinic with complaints of bilateral blurred vision 10 days following the first dose of inactivated vaccine (Sinopharm) and again got severe visual loss in the right eye 2 days after the second dose of vaccination (Sinovac). On examination, the visual acuity was hand motion OD, and 0.25 OS and the intraocular pressures were 11.2 mmHg OD and 11.3 mmHg OS. An anterior segment examination revealed a conjunctiva congestion, gray fine keratic precipitate, and anterior chamber cell in both eyes. The dilated fundus examination showed multiple serous retinal detachment (Figure 2), which was consistent with intraretinal and subretinal fluid, shown on OCT Figure 2B. Tests for HIV, syphilis and tuberculosis were negative. The patient was diagnosed with VKH, and systemic steroids were initiated. Visual acuity improved to 1.0 in both eyes with resolved subretinal detachment (Figure 2C,D).

Case 3

A previously healthy 36-year-old female presented to the retina clinic with sudden painless vision loss in her left eye one week after receiving the first inactivated COVID-19 [12] vaccine (Sinopharm). She had also experienced mild flu-like symptoms 3 days after the vaccination. Past ocular and medical history were unremarkable. An elevated erythrocyte sedimentation rate (ESR) of 41.8 mm/h was noted, while the systemic workup had ruled out infection. Visual acuity was 0.2 OD and 1.0 OS with normal intraocular pressure. The fundoscopy of the right eye showed a subretinal foveal yellowish-white lesion (Figure 3A), corresponding to the disruption of the ellipsoid layer and retinal pigment epithelium (RPE) as shown in OCT (Figure 3B). FA showed foveal hyperfluorescent from staining in the late phase (Figure 3C–E). The changes in the retina were consistent with AIM, and oral steroids with a tapering dose were initiated. Visual acuity improved to 0.6 OD with resolution of RPE lesion at a 4-week follow-up (Figure 3F).

## 4. Discussion

In the current case series, we reported 10 eyes from seven patients presenting with ocular adverse events shortly after the first inoculation with an inactivated COVID-19 vaccine (Sinopharm or Sinovac).

In this set of patients, ocular manifestation included VKH-like uveitis, choroiditis, iritis, and AIM. AIM was previously reported following coxsackievirus infection and H1N1 vaccination, usually preceded by flu-like symptoms [13]. As far as we know, this is the first reported inactivated COVID-19 vaccine-related AIM.

Despite the large number of vaccine doses being administered worldwide, little is known about their adverse events and mechanism of inducing different adverse events profiles [14]. Both vaccines are produced by Vero cells which are infected with COVID-19 virus. After culturing the cells, the virus is harvested, inactivated, concentrated, purified, and mixed with an adjuvant (aluminum hydroxide) to enhance the immunogenicity. Inactivated vaccines provoke an immune response by humoral immunity, therefore, they require multiple doses [15]. A third booster is encouraged in China to induce a stronger protection [16]. Commonly proposed mechanisms of inflammation may be induced by the adjuvants and have been classified as Shoenfeld syndrome typically occurring in patients with an autoimmune disorder, often accompanied by arthralgia and fatigue [17].

Recently, a case of AIM following COVID-19 infection was reported by Venkatesh and associates [18]. AIM following the Pfizer mRNA COVID-19 vaccine was reported as well [19]. Although AIM with prodromal flu-like symptoms resembles a direct viral infection, the molecular mimicry between a spike protein and a component of the RPE cells may lead to the activation of the host immune response.

VKH or VKH-like disease has previously been reported with inactivated vaccines against yellow fever, influenza, and HPV, which may have been induced by the adjuvants [20,21]. While there are reported new onset and reactivation of VKH disease related to the Pfizer mRNA vaccine, in view of the similar phenotype of uveitis despite a different type of vaccine, the antigen mimicry from the virus instead could be a potential trigger [12,22,23].

Only two patients received the second dose of the vaccine. One patient (Case 2) diagnosed with VKH had blurred vision 10 days following the first dose of Sinopharm vaccine and complained of much more severe vision loss 2 days after the second dose of Sinovac vaccine, which was consistent with bullous retinal detachment. This patient did not receive treatment prior to the second vaccination. We highly suspected that the vaccine was the causative factor due to the two similar reactions to the same drugs. Interestingly, this patient received mixed vaccines, which may induce much stronger immunogenicity, thus producing more severe adverse reactions after the second vaccination [2].

By contrast, another patient diagnosed with multifocal choroiditis after the first Sinovac vaccine was timely treated with periocular steroids. When inoculated with the second dose of Sinovac vaccine, no ocular symptoms were noted. In regard to this patient, the timely detection and treatment may have led to a favorable outcome.

Additionally, the aqueous humor analysis in three patients (Cases 1, 3 and 4) showed significant elevated IL-6 and moderately elevated INF-γ levels. IL-4 and IL-10 were unremarkable. More laboratory tests are needed to elucidate the mechanism of vaccine-related ocular inflammation.

The main limitation of our study, was its retrospective design. The establishment of a causal relationship between the vaccine and the observed ocular diseases is challenging since vaccines are delivered in huge numbers. Although a thorough workup was done for our patients, it is hard to rule out coincidence for rare events.

Nevertheless, potential ocular adverse events following the administration of inactivated COVID-19 vaccines are encouraged to be reported with detailed information to increase the awareness of the health community for timely detection and proper treatment.

## Figures and Tables

**Figure 1 vaccines-10-00482-f001:**
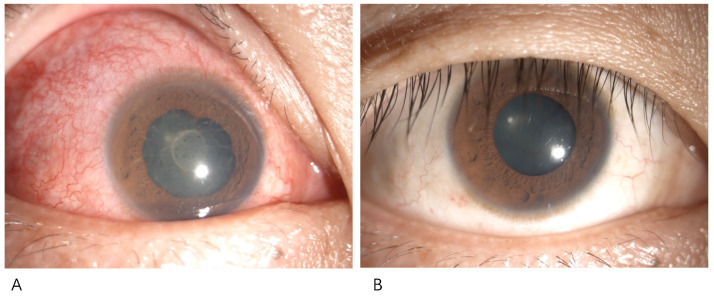
Iritis after administration of inactivated COVID-19 vaccine. The right eye of a patient presented with iritis 3 days following the first dose of inactivated COVID-19 vaccine (**A**). The patient was treated with topical and periocular steroid and the inflammation resolved at a 2-week follow-up (**B**).

**Figure 2 vaccines-10-00482-f002:**
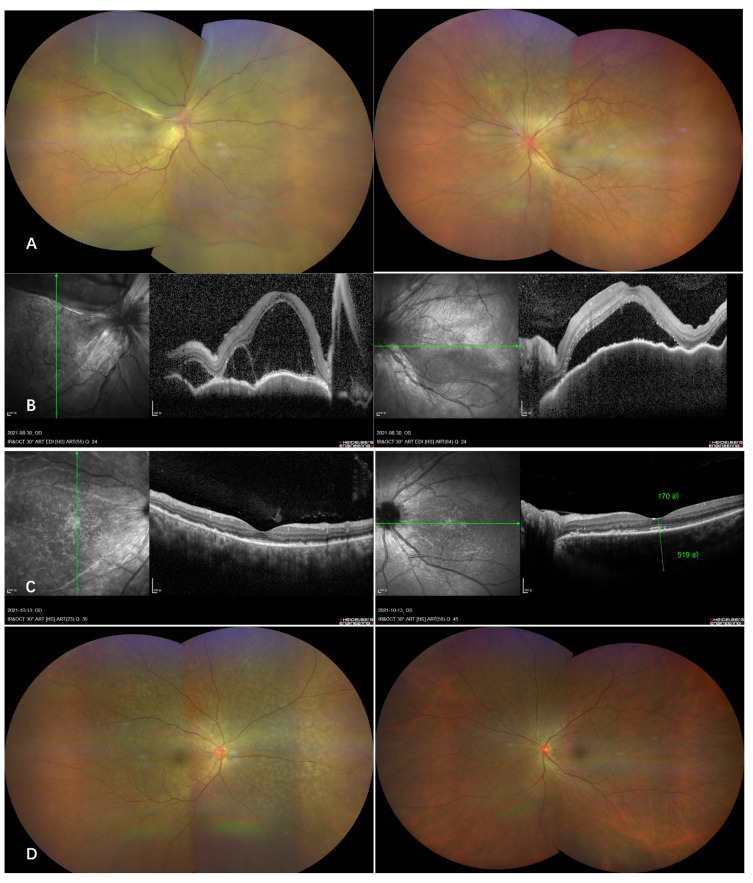
Acute onset of Vogt–Koyanagi–Harada disease after administration of inactivated COVID-19 vaccine. Wide-field color fundus photography of a previous healthy woman who developed bilateral multiple serous retinal detachment 10 days following the first dose of the Sinopharm inactivated COVID-19 vaccine (**A**), corresponding to the OCT revealing intraretinal and subretinal fluid (**B**). The patient started on oral steroid with tapering dose, and the bullous retinal detachment resolved completely after 6 weeks of follow-up (**C** and **D**).

**Figure 3 vaccines-10-00482-f003:**
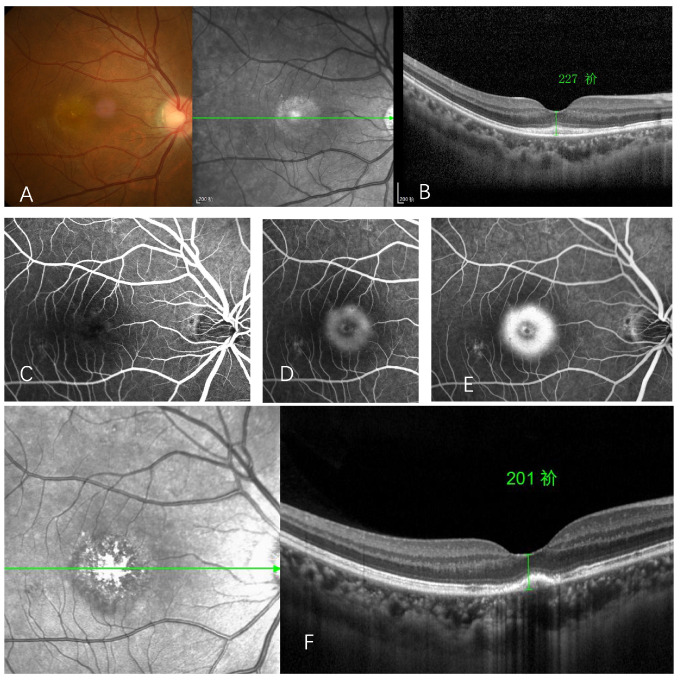
Acute idiopahtic maculopathy following inactivated COVID-19 vaccine administration. Fundus photography of a patient presented with sudden painless visual loss in the right eye one week following the first dose of inactivated COVID-19 vaccine showing subretinal foveal yellowish-white lesion (**A**), corresponding to hyperplasia of the retinal pigment epithelium (RPE) and disruption of the ellipsoid layer showed in OCT (**B**), and late staining in the late phase of FA (**C**–**E**). Resolution of RPE hyperplasia was noted at the 4-week follow-up (**F**).

**Table 1 vaccines-10-00482-t001:** Patients’ demographics, medical history and symptoms post inactivated COVID-19 vaccine.

Patient No.	Age (Years)	Gender	Systemic and Ocular History	Medication	Symptoms after the First Dose of Vaccination	Vaccination Site (City)	First Dose of Vaccine Received	Second Dose of Vaccine Received
1	33	M	Ankylosing Spondylitis	Etanercept	No	Zhangzhou	Sinopharm *	No
2	57	F	No	No	No	Zhangzhou	Sinopharm *	Sinovac
3	21	M	No	No	fatigue, headache	Putian	Sinovac	No
4	30	F	No	No	No	Putian	Sinovac	Sinovac
5	36	F	No	No	No	Sanming	Sinovac	No
6	28	F	No	No	flu-like symptoms	Quanzhou	Sinopharm	No
7	10	F	Intermediate uveitis	Adalimumab methotrexate	No	Zhangzhou	Sinovac	No

M = male, F = Female. * Sinopharm and Sinovac are both inactivated COVID-19 vaccines.

**Table 2 vaccines-10-00482-t002:** Ocular disease after receiving the inactivated COVID-19 vaccine.

Patient No.	Eyes Involved	Symptoms after 1st Vaccine	Symptoms after 2nd Vaccine	Manifestation	Time Intervals between Vaccination and Symptoms Onset (Days)	Positive Serologic Test	Treatment Received	Outcome	Causality Assessment
1	Unilateral	Redness, blurred vision	-	Iritis	3	HLA-B27+; ESR 38.0 mm/h	Periocular steroids	CR	Probable
2	Bilateral	Redness, blurred vision	Deteriorated vision	VKH	10	-	Oral steroids	CR	Probable
3	Bilateral	Redness, blurred vision	-	VKH-like uveitis	1	-	Periocular steroids	CR	Probable
4	Unilateral	Blurred vision	No symptoms	Multifocal choroiditis	3	-	Periocular steroids	CR	Possible
5	Unilateral	Blurred vision	-	AIM	7	ESR 41.8 mm/h	Oral steroids	CR	Possible
6	Bilateral	Blurred vision	-	VKH-like uveitis	3	-	Oral steroids	CR	Probable
7	Unilateral	Redness	-	Episcleritis	7	-	Topical steroid	CR	Possible

VKH = Vogt–Koyanagi–Harada, AIM = acute idiopathic maculopathy, HLA = human leukocyte antigen, ESR = erythrocyte sedimentation rate, CR = complete remission.

## Data Availability

Please refer to suggested Data Availability Statements in section “MDPI Research Data Policies” at https://www.mdpi.com/ethics (accessed on 16 March 2022).

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
