# Peer review of "Ocular Adverse Events after Inactivated COVID-19 Vaccination in Xiamen"

_vaccines, 2022, doi:10.3390/vaccines10030482_

Round 1
Reviewer 1 Report
The article presents a series of ocular AE cases without specifying the time interval in which they were observed, the number of people vaccinated in a given area, and a comparison with the number of similar diagnoses in the unvaccinated population. The authors did not prove a causal relationship with vaccination, yet the conclusions (included in the abstract) state the existence of such a relationship. At the same time, the Discussion does not provide sufficient premises for drawing such conclusions.
Author Response
Response to Reviewer 1 Comments
Dear reviewer,
On behalf of all co-authors, I would like to thank you for your constructive comments, which have helped us to considerably improve the paper. Please see our point-by-point response (in blue) to the comments below.
Yours sincerely,
Xiuju Chen
General comments: The article presents a series of ocular AE cases without specifying the time interval in which they were observed, the number of people vaccinated in a given area, and a comparison with the number of similar diagnoses in the unvaccinated population. The authors did not prove a causal relationship with vaccination, yet the conclusions (included in the abstract) state the existence of such a relationship. At the same time, the Discussion does not provide sufficient premises for drawing such conclusions.
Response: Thank you for your concise and constructive comments. We are encouraged to study more literature for better interpreting our work. In order to explain more clearly, we divided the comments to 2 points below.
Point 1: The article presents a series of ocular AE cases without specifying the time interval in which they were observed, the number of people vaccinated in a given area, and a comparison with the number of similar diagnoses in the unvaccinated population.
Response 1: Thank you for your precious comments and advice. This case series took place at a tertiary referral center in the southeast of China (Xiamen Eye Center in Fujian Province) from February 2021 to July 2021. A period that all individuals are encouraged to have COVID-19 vaccine.
As of August 2021, a total of 49.30 million vaccines has been used across the province, and 30.34 million people (age range: 12-60 years) has received the 1st dose of vaccine, accounting for 73% of the residents. (Data from COVID-19 press conference by Fujian Government, August 18th, 2021).
As of 14th March 2022, a total of 94.6331 million vaccines has been used across the province, and 38.7991 million people (age range: 12-60 years) has received the 1st dose of vaccine. In addition, 18.4154 million people received booster (the 3rd dose) immunizations. (Center for Disease Control, Fujian Province, data cited 15th March,2022).
An invited commentary by Jampol LM, JAMA 2021 calculated the incidence of cerebral venous sinus thrombosis (CVST) in individuals vaccinated with vector vaccine (5 cases per million doses) and mRNA COVID-19 vaccine respectively (0.87 cases per million doses). A causal relationship with vector-based vaccination thus is considered plausible by regulatory agencies.
It is much convincing if we could compare the reporting rate of ocular events in people vaccinated in a given area, with the incidence of similar diagnoses in the unvaccinated population. We acknowledged the limitation of our study. It is a mono center retrospective case series with a spectrum of retinal events, and not all of the patients who were diagnosed with vaccine-related ocular disease were referred to Xiamen Eye Center. The reporting rate of ocular AE is not available at this time but to add one column ‘vaccination site’ in table 1 to provide the exact area the patients coming from.
Point 2: The authors did not prove a causal relationship with vaccination, yet the conclusions (included in the abstract) state the existence of such a relationship. At the same time, the Discussion does not provide sufficient premises for drawing such conclusions.
Response 2: Thank you for pointing out this essential flaw in our previous manuscript. We have made an extensive revision to the abstract, method and discussion. We are grateful if you can go through the revised manuscript and looking forward to your further comments and suggestion. Here we briefly list the main revisions below:
Abstract:
- we have amended the conclusion to ‘Potential ocular adverse events should be reported to increase the awareness of the health community for timely detection and proper treatment.’
Methods:
- The establishment of a causal relationship between the ocular events and vaccine is be guided WHO Adverse Drug Terminology and Naranjo Criteria.
- A thorough workup to exclude other cause of ocular disease is present.
Results:
- The patients’ medication, systemic symptoms, positive results in workup, vaccination site and causality assessment were listed in table 1 and table 2.
- To describe two cases who received the 2nd dose of vaccine were clearly described. One cases with VKH had exacerbation of visual loss following 2nd
vaccination without prior treatment. This is informative since a booster dose is encouraged in China.
Discussion:
- Update the adverse events with newly published literature. Especially in Acute idiopathic maculopathy, which was reporting following COVID-19 infection and Pfizer mRNA COVID-19 vaccine.
- The components of Inactivated vaccine as well as proposed mechanism of vaccine induced ocular inflammation were described.
- The impact of the 2nd dose of vaccine is addressed.
- To acknowledge the limitation of our study.
Reviewer 2 Report
Known in the field based on previous literatures:
- Covid-19 is an infectious disease caused by severe acute respiratory syndrome coronavirus 2 (SARS-CoV-2). The disease has blowout worldwide, leading to an ongoing pandemic.
- Symptoms of COVID‑19 are variable, but often include cough, breathing difficulties, headache, fever, fatigue, loss of smell and taste.
- COVID‑19 vaccine is developed and intended to provide acquired immunity against the virus.
In this review authors reported following findings:
- Although, Covid-19 vaccine is effective, but several side effects were speculated. In this manuscript, authors reported another side effect on eye which is very uncommonly notice in world- the ocular adverse effects.
- Authors studied series of ocular adverse events in cases who were received 1st or 2nd dose of inactivated Covid-19 vaccine.
- Authors reported significant elevated IL-6 with negative COVID-19 virus copy in aqueous humor in 3 patients (Case 1, 3 and case 4).
Authors nicely mentioned many facts related to ocular adverse effects in person who received 1st or 2nd dose of Covid-19 vaccine. Although, we need some more experimental evidence as well as more cases to establish the role of Covid-19 vaccine. The data provided by authors based on few cases which is insufficient, and we cannot generalize but localized. The following suggestions if incorporated could help in the better understanding of the significance of the work and implications.
Minor/Major Concerns:
- In line 31, after examination it should be coma (,) rather than stop (.) sign.
- Explain why you measured only one inflammatory cytokine- IL-6 but what about other inflammatory cytokines.
- What was the other side effects after 1st or 2nd dose of inactivated Covid-19 vaccine on ocular history cases? Add a table listing the different side effects.
Author Response
Response to Reviewer 2 Comments
Dear reviewer,
On behalf of all co-authors, I would like to thank you for your careful and
constructive comments, which have helped us to considerably improve the paper. Please see our point-by-point response (in blue) to the comments below.
Yours sincerely,
Xiuju Chen
Key points:
known in the field based on previous literatures:
- Covid-19 is an infectious disease caused by severe acute respiratory syndrome coronavirus 2 (SARS-CoV-2). The disease has blowout worldwide, leading to an ongoing pandemic.
- Symptoms of COVID‑19 are variable, but often include cough, breathing difficulties, headache, fever, fatigue, loss of smell and taste.
- COVID‑19 vaccine is developed and intended to provide acquired immunity against the virus.
In this review authors reported following findings:
- Although, Covid-19 vaccine is effective, but several side effects were speculated. In this manuscript, authors reported another side effect on eye which is very uncommonly notice in world- the ocular adverse effects.
- Authors studied series of ocular adverse events in cases who were received 1stor 2nd dose of inactivated Covid-19 vaccine.
- Authors reported significant elevated IL-6 with negative COVID-19 virus copy in aqueous humor in 3 patients (Case 1, 3 and case 4).
Response: Thank you for reviewer’s concise and brief key points for our manuscript.
General comments: Authors nicely mentioned many facts related to ocular adverse effects in person who received 1stor 2nd dose of Covid-19 vaccine. Although, we need some more experimental evidence as well as more cases to establish the role of Covid-19 vaccine. The data provided by authors based on few cases which is insufficient, and we cannot generalize but localized. The following suggestions if incorporated could help in the better understanding of the significance of the work and implications.
Response: We are extremely grateful to reviewer for suggestions on how to
better address the significance of the work and implications. We have made an extensive revision to the abstract, method and discussion. We are grateful if you can go through the revised manuscript and looking forward to your further comments and suggestion.
Minor/Major Concerns:
Point 1: In line 31, after examination it should be coma (,) rather than stop (.) sign.
Response 1: We are sorry for the mistake. It should be corrected.
Point 2: Explain why you measured only one inflammatory cytokine- IL-6 but what about other inflammatory cytokines.
Response 2: Thank the reviewer for underlying the deficiency.IL-6 is a pro-inflammatory cytokine that usually noted elevated in eyes with inflammation. But a single cytokine could not give the sufficient information. A revision has been done in
The following parts:
Methods (Line 68-71):
Intraocular fluid was sent to detect infection, including COVID- 19 virus, toxoplasmosis (TOXO), herpes virus (HSV), varicella-zoster virus (VZV), cytomegalovirus (CMV), and proinflammatory cytokines, including interleukin (IL)-2, IL-4, IL-6, IL-8, IL-10, IL-17, IFN-Y, IFN-a, and TNF-a (Qingdao Raisecare Biotech) if applicable.
Results (Line 101-105):
Aqueous humor was analyzed in 3 selected patients (Case 1 with iritis, Case 3 and Case 4 with VKH-like disease) and revealed significantly elevated IL-6 (range 734.2 pg/ml ∼ 2120.4 pg/ml) and negativemild to moderate elevated IFN-Y (range 27.4 pg/ml ∼ 855.0 pg/ml). IL-4 and IL-10 were within normal limits. Results of COVID-19 virus copy, CMV, HSV, VZV and TORCH IgM were negative.
Discussions (Line 195-198):
Additionally, aqueous humor analysis in three patients (case 1, 3 and case 4) showed significant elevated IL-6 and moderate elevated INF-Y levels. IL-4 and IL-10 are unremarkable. More laboratory tests are needed to elucidate the mechanism of vaccine-related ocular inflammation.
、
Point 3. What was the other side effects after 1st or 2nd dose of inactivated Covid-19 vaccine on ocular history cases? Add a table listing the different side effects.
Response 3: We are grateful for the suggestion. All the information is well placed in table 1 and table 2 (Line 91-92), including other side effects, medication, symptoms post 1st or 2nd dose of vaccine, vaccinate site, positive serologic tests, causality assessment. We aimed to provide more details on the case series for better understanding the ocular adverse events.
Reviewer 3 Report
The manuscript is well-organized but I have detected some English composition mistakes (i.e. -Line 8: “were obtained”, obtained not obtain; line 116, “previous reported”, previously reported), so, if possible, a detailed copyediting must be done by an English-native speaker.
My main comments are concerning Methods (must be more detailed and explained) and Discussion (would improve with a longer discussion, referring other evidences). That is, authors must refer other similar studies like those by Rabinovitch et al., Retina. 2021, and Renisi et al., Int J Infect Dis. 2021).
Other minor comments are:
-Literature is updated and properly cited. However, I have some comments regarding reference 2. Differently from the rest of references (even when the number of authors is higher than 6 like #3 and #4), the list of authors is not depicted, just abbreviated with “et al.”
-Were medications collected from the patients included? They must show this datum in a Table (like Table 1) as well as medical and ocular history for each patient.
-A second dose of vaccine is mentioned for two patients. Authors must clarify if uveitis appeared after the first dose or after the second one. This is confusing. In addition, the time interval between vaccination (first and second doses) and initiation of symptoms should also be mentioned for these two patients. Moreover, the contribution or not of a second dose must be discussed in Discussion. Is there any evidence of this?
-Lines 67-69. They mention certain viruses but I suppose that bacteria like Treponema pallidum and Mycobacterium tuberculosis were also discarded. They should mention the specific tests performed in all the patients in Methods (not only the microbiological ones but also biochemical and clinical assays).
Author Response
Response to Reviewer 3 Comments
Dear reviewer,
On behalf of all co-authors, I would like to thank you for your careful and
constructive comments, which have helped us to considerably improve the paper. Please see our point-by-point response (in blue) to the comments below.
Yours sincerely,
Xiuju Chen
English language and style: The manuscript is well-organized, but I have detected some English composition mistakes (i.e. -Line 8: “were obtained”, obtained not obtain; line 116, “previous reported”, previously reported), so, if possible, a detailed copyediting must be done by an English-native speaker.
Response: Thank you for your positive words of our work. We are apologized for the grammar mistakes in the original manuscript. The language in revision manuscript was improved with assistance from a native English speaker.
General comments: My main comments are concerning Methods (must be more detailed and explained) and Discussion (would improve with a longer discussion, referring other evidence). That is, authors must refer other similar studies like those by Rabinovitch et al., Retina. 2021, and Renisi et al., Int J Infect Dis. 2021).
Response: Thank you for the constructive suggestions! After learning from Rabinovitch and Renisi’ study, much more information is present in the revised manuscript for better understanding of our work. We have added a detailed description as follows:
Methods (Line 63-71)
Accessory tests were performed with aims to exclude other causes of ocular events. These included complete blood count, biochemical analysis, erythrocyte sedimentation rate (ESR), C-reactive protein (CRP), rheumatoid factor (RF), antinuclear antibodies (ANA), Human leukocyte antigen (HLA)-B27. Serologic screening for TORCH, HIV, syphilis (treponema pallidum hemagglutination) and tuberculosis (purified protein derivative) were performed in all cases. Intraocular fluid was sent to detect infection, including COVID19 virus, toxoplasmosis (TOXO), herpes virus (HSV), varicella-zoster virus (VZV), cytomegalovirus (CMV), and proinflammatory cytokines, including interleukin (IL)-2, IL-4, IL-6, IL-8, IL-10, IL-17, IFN-Y, IFN-a, and TNF-a (Qingdao Raisecare Biotech) if applicable.
Discussions:
- Update the adverse events with newly published literature. Especially in Acute idiopathic maculopathy, which was reporting following COVID-19 infection and Pfizer mRNA COVID-19 vaccine. (Line 170-174)
- The components of Inactivated vaccine as well as proposed mechanism of vaccine induced ocular inflammation were described. (Line 162-169)
- The impact of the 2nd dose of vaccine is addressed. (Line 180-194)
- To acknowledge the limitation of our study. (Line 199-202)
Other minor comments:
Point 1: Literature is updated and properly cited. However, I have some comments regarding reference 2. Differently from the rest of references (even when the number of authors is higher than 6 like #3 and #4), the list of authors is not depicted, just abbreviated with “et al.”
Response: Thank you for pointing out this bug. We have corrected. Meanwhile, the literature is undated and properly cited. We are appreciated for all the studies cited that help to improve the manuscript.
-Were medications collected from the patients included? They must show this datum in a Table (like Table 1) as well as medical and ocular history for each patient.
Response: Thank you for your kind reminders, the medication, medical and ocular history is well described in revised table 1 (Space between line 91-91). In addition, much more information is added in table 2 (Space between line 91-91in attempt to clearly present the cases.
-A second dose of vaccine is mentioned for two patients. Authors must clarify if uveitis appeared after the first dose or after the second one. This is confusing. In addition, the time interval between vaccination (first and second doses) and initiation of symptoms should also be mentioned for these two patients. Moreover, the contribution or not of a second dose must be discussed in Discussion. Is there any evidence of this?
Response: Thanks for underlining this deficiency. This section was revised and modified according to the reviewer’s suggestions. They have been shown in the manuscript as follows:
Results (Line 92-100)
Only two patients received the 2nd dose of the vaccine. One patient (Case2, see more information in selected cases below) who had a decreased vision 10 days after the 1st dose of Sinopharm vaccine, had exacerbation of vision loss to hand motion in the right eye (OD) 2 days following the 2nd dose of Sinovac vaccine. Prior to the 2nd dose vaccination, The patient did not receive any treatment. The patient was diagnosed with bilateral VKH and treated with oral steroids. The vision recovered to 1.0 OD in a 4-week follow-up. and 1 One patient (Case 4) diagnosed with multifocal choroiditis in the left eye (OS) after the 1st dose of Sinovac vaccine immediately received periocular triamcinolone acetonide 40mg OS. The patient got the 2nd dose of Sinovac vaccine one month later without visual deterioration.
Discussion (Line 180-207)
Only two patients received the second dose of the vaccine. One patient (case 2) diagnosed with VKH had blurred vision 10 days following the 1st dose of Sinopharm vaccine and complaint of much more severe vision loss 2 days after the 2nd dose of Sinovac vaccine, which was consistent with bullous retinal detachment. This patient did not receive treatment prior to the 2nd vaccination. We highly suspected that the vaccine was the causative factor due to the two similar reactions to the same drugs. Interestingly, this patient received mixing vaccines, which may induce much stronger immunogenicity, thus producing more severe adverse reactions after the 2nd vaccination.
By contrast, another patient diagnosed with multifocal choroiditis after the 1st Sinovac vaccine was timely treated with periocular steroids. When inoculated with the 2nd dose of Sinovac vaccine, no ocular symptoms were noted. In regard to this patient, timely detection and treatment may lead to a favorable outcome.
-Lines 67-69. They mention certain viruses, but I suppose that bacteria like Treponema pallidum and Mycobacterium tuberculosis were also discarded. They should mention the specific tests performed in all the patients in Methods (not only the microbiological ones but also biochemical and clinical assays).
Response: Thank you for nicely mention about the other infections. The laboratory tests were listed in detail in Methods ((Line 69-71) including aqueous humor analysis.
Round 2
Reviewer 1 Report
Can be published in the current version